# Dietary Supplementation with *Bupleuri Radix* Reduces Oxidative Stress Occurring during Growth by Regulating Rumen Microbes and Metabolites

**DOI:** 10.3390/ani14060927

**Published:** 2024-03-17

**Authors:** Cheng Pan, Haiyan Li, Fuqiang Wang, Jianping Qin, Yanping Huang, Wangsheng Zhao

**Affiliations:** 1School of Life Science and Engineering, Southwest University of Science and Technology, Mianyang 621000, China; panchengswust@163.com (C.P.); 17378586321@163.com (H.L.); 18881690362@163.com (F.W.); 2Shenmu Livestock Development Center, Yulin 719000, China; 18729940010@163.com

**Keywords:** oxidative stress, total antioxidant capacity, Shanbei Fine-Wool Sheep, *Bupleuri Radix*, multiomics

## Abstract

**Simple Summary:**

Oxidative stress is closely related to animal health. *Bupleuri Radix*, a well-known Chinese herb, has an important antioxidant capacity. In this study, we discovered that supplementing *Bupleuri Radix* to the diet could reduce oxidative stress during the growth stage of Shanbei fine-wool sheep. In addition, we analyzed the antioxidant mechanism of *Bupleuri Radix* by combining metabolomics and microbiomics and discovered that *Bupleuri Radix* may increase the antioxidant capacity of Shanbei fine-wool sheep by regulating rumen bacteria and metabolites. The findings indicate that implementing appropriate nutrition strategies can improve the health of Shanbei fine-wool sheep.

**Abstract:**

Oxidative stress (OS) in ruminants is closely associated with disease; thus, improving antioxidant capacity is an important strategy for maintaining host health. *Bupleuri Radix* (BR) could significantly improve host health and stress levels. However, the clear antioxidant mechanism of the function of BR remains unknown. In the current study, LC-MS metabolomics combined with 16S rRNA gene sequencing was employed to explore the effects of BR on rumen microbiota and metabolites in Shanbei Fine-Wool Sheep (SFWS), and Spearman correlation analyses of rumen microbiota, metabolites, and OS were performed to investigate the mechanism of antioxidant function of BR. Our results indicated that as SFWS grows, levels of OS and antioxidant capacity increase dramatically, but providing BR to SFWS enhances antioxidant capacity while decreasing OS. Rumen microbiota and OS are strongly correlated, with total antioxidant capacity (T-AOC) showing a significant negative correlation with *Succiniclasticum* and a positive correlation with *Ruminococcus*. Importantly, the Chao1 index was significantly negatively correlated with malondialdehyde (MDA) and positively correlated with superoxide dismutase (SOD) and T-AOC. Two biomarkers connected to the antioxidant effects of BR, 5,6-DHET and LPA (a-25:0/0:0), were screened according to the results of metabolomics and Spearman analysis of rumen contents, and a significant relationship between the concentration of rumen metabolites and OS was found. Five metabolic pathways, including glycerolipid, glutathione, nucleotide, D-amino acid, and inositol phosphate metabolism, may have a role in OS. The integrated results indicate that rumen microbiota and metabolites are strongly related to OS and that BR is responsible for reducing OS and improving antioxidant capacity in post-weaned SFWS. These findings provide new strategies to reduce OS occurring during SFWS growth.

## 1. Introduction

The contributions of oxidative stress (OS) to animal growth and development, energy balance, metabolism, diets, production traits, and health conditions have been widely studied [1,2,3,4,5]. Animals have to meet the metabolic demands imposed by physiological changes during various periods of their lives, triggering a large increase in oxidative metabolism and an increase in the number of free radicals in organs and cells [6]. Free radicals are a byproduct of the body’s normal metabolic activities [7]. OS is caused by an overabundance of free radicals, a lack of antioxidant capacity, or a combination of both. Ruminant illnesses, including sepsis, mastitis, acidosis, ketosis, enteritis, and pneumonia, have been related to OS [8,9,10]. Thus, one of the most significant strategies for keeping ruminants healthy is to increase antioxidant capacity.

As a traditional Chinese medicine, *Bupleuri Radix* (BR) contains functional components such as saponins, flavonoids, and polysaccharides, which may regulate several signaling pathways providing antioxidant, antitumor, anticancer, antibacterial, and immunomodulatory effects [11,12,13]. BR significantly reduces reactive oxygen species (ROS) and lipid peroxidation products (MDA), while also increasing the activity of antioxidant enzymes such as superoxide dismutase (SOD), catalase (CAT), and glutathione peroxidase (GSH-Px) [13]. Saponin A isolated from BR cured smoking-induced pneumonia in mice by greatly improving the antioxidant effect in lung tissues while lowering myeloperoxidase and MDA [14]. Currently, the antioxidant impact of BR has been widely used in animals such as mice [15], tilapia [16], and broiler chickens [17]. However, the antioxidant effects of BR in ruminants have received less attention, particularly in Shanbei Fine-Wool Sheep (SFWS) health, growth, and development.

The four complicated stomachs in ruminants—the reticulum, abomasum, omasum, and rumen—have an exceptionally complex microbiota, and the rumen bacteria in these stomachs are crucial to preserving the host organism’s homeostasis [18,19,20]. Rumen microbiota and OS are related, and this connection is crucial for preserving immune system performance, nutritional absorption, and metabolic dynamic balance [21]. Alterations in the ruminant gastrointestinal bacteria owing to factors like weaning, high temperatures, and aging may also result in changes in microbial metabolism and microbial enzyme activities. These changes could lead to disruptions in the metabolism of nutrients, imbalances in the homeostasis of free radicals, and oxidative damage to cells and mitochondria [21,22,23,24,25]. The research mentioned above indicates that rumen microbes and their metabolites are important mediators in the host OS process; however, it is yet unknown how both of these interact with OS.

This study aimed to evaluate the impact of BR on the diversity and composition of the rumen microbiota, enzyme activities, metabolites, and OS. Additionally, it sought to investigate how rumen microbiota and metabolites interact to improve the antioxidant effect. After the SFWS was weaned, the effects of various BR percentages on the rumen microbiota, enzyme activity, metabolites, and OS were assessed. Microbial and metabolic alterations in the rumen were evaluated using metabolomics analysis and microbiome analysis. The best BR % feeding schedule for enhancing antioxidant resistance in post-weaning SFWS was found, providing the way for a rumen microbiota-shaping strategy to enhance SFWS health.

## 2. Materials and Methods

### 2.1. Ethics Statement

The protocols for collecting samples were approved by the Southwest University of Science and Technology’s Livestock and Poultry Breeding Professional Committee, and all studies involving animals were carried out in compliance with the Southwest University of Science and Technology (Approval No. L2022026-2022-03).

### 2.2. Experimental Design and Sample Collection

The test animals were procured from the original SWFS breeding farm located in Shenmu City, Shaanxi Province. Forty SWFS (50 percent male and 50 percent female) that were robust, had been weaned at three months of age, and had similar body weights were picked. Ten animals per group, five males and five females, were randomly assigned to one of four groups: control (fed basal diet); treatment groups (1%, 2%, and 4% BR, respectively); all groups received freely provided watering and foraging in separate enclosures (3 m^2^). The composition and nutrient levels of the basal diet are shown in Table 1. Every day at 8:00 a.m. and 6:00 p.m., feedings were performed. The main test period lasted 60 days, while the pre-test period lasted seven days, for a total of 67 days for the experimental period. The branches of the *Bupleuri Radix* by-products are the primary source of the BR powder. On the first day of the experiment, the experimental animals’ body weights were recorded, and blood was collected from the jugular vein. One day before the experiment’s end, all test animals were fasted. Then, rumen fluid was extracted using a rumen fluid extractor, and jugular vein blood was drawn. The animals were all weighed and recorded. The rumen fluid was collected, filtered through four layers of gauze, and then put into 2 mL frozen storage tubes and kept in liquid nitrogen to extract microbial DNA and measure the activity of rumen enzymes. Serum was extracted from samples by centrifuging them at 3000× *g* for 15 min at 4 °C right after each blood collection. For further investigation, the serum was kept at −80 °C after being frozen in liquid nitrogen.

### 2.3. Measurement of Serum Oxidative Parameters

Serum CAT, MDA, SOD, and T-AOC were determined using commercial assay kits from Nanjing Jiancheng Bioengineering Institute (Nanjing, China) by the manufacturer’s instructions.

### 2.4. Measurement of Rumen Enzyme Activity

To measure the rumen fluid enzyme activities, a portion of the collected rumen fluid was frozen at 4 °C for each sheep. The supernatant was then extracted by centrifugation at 3500 r/min for 10 min at 4 °C after thawing. The assay was conducted using the Nanjing Jianjian Reagent Kit’s provided assay method. The absorbance value of the samples was then determined using an enzyme marker, and the activities of pepsin, lipases, α-amylase, and cellulase were computed based on the formulas given in their corresponding descriptions.

### 2.5. Microbiome Sample Processing and Sequencing

Total genome DNA was extracted from the rumen fluid of SFWS using the CTAB method. The concentration and purity of DNA were measured on a 1% agarose gel. Based on the concentration, DNA was diluted to 1 ng/L using sterile water. The PCR procedures included 15 L of Phusion^®^ High-Fidelity PCR Master Mix (New England Biolabs, Ipswich, MA, USA), 2 mM of forward and reverse primers, and around 10 ng of template DNA. A one-minute initial denaturation at 98 °C was followed by thirty cycles of ten seconds at 98 °C denaturation, thirty seconds of annealing at 50 °C, and thirty seconds of elongation at 72 °C throughout the thermal cycling process, and lastly, five minutes at 72 °C. We combined the PCR products with an equal volume of 1× loading buffer that contains SYB green and then ran the electrophoresis on a 2% agarose gel to detect the results. Equidensity ratios were used to combine the PCR products. A mixture of PCR products was then purified using a Qiagen Gel Extraction Kit (Qiagen, Germany). Following the creation of sequencing libraries using the TruSeq^®^ DNA PCR-Free Sample Preparation Kit (Illumina, San Diego, CA, USA) in line with the manufacturer’s instructions, index codes were added. The library’s quality was assessed using the Agilent Bioanalyzer 2100 instrument and the Qubit@ 2.0 Fluorometer (Thermo Scientific, Waltham, MA, USA). After this, the library was sequenced using an Illumina NovaSeq machine, producing 250 bp paired-end reads.

### 2.6. Bacterial Metagenome Bioinformatics and Statistical Analysis

By carrying out the subsequent preprocedures, the raw reads from high-throughput sequencing were quality-filtered to produce effective reads. Primer sequences were trimmed to obtain clean reads using cutadapt 1.9.1 software, and then UCHIME v4.2 software was used to identify and remove chimera sequences. Additionally, using Quantitative Insights into Microbial Ecology (QIIME) software (Uparse v7.0.1001), the effective sequence alignment carried out using the SILVA database was clustered into operational taxonomic units (OTUs) with a threshold of 97% sequence similarity. The taxonomic assignment of all sequences and microbial composition were analyzed based on normalized output data. The calculations of alpha diversity (Chao1, ACE, Shannon, and PD_whole_tree) were performed in QIIME software(1.8.0). Using the R program (v3.6.0), the rarefaction curve, Shannon curve, rank abundance curve, species accumulation curve, and Good’s coverage were displayed to show species abundance and evenness as well as whether the present sequencing depth has captured the great majority of species information. Principal coordinates analysis (PCoA) and the unweighted pair group technique with arithmetic mean (UPGMA) studied the similarities between groups or individuals. To identify biomarkers between sample groups, metastatic analyses (*t*-test) were carried out at several taxonomic levels (phylum and genus) to evaluate the variation in the relative abundance of microbiome members.

### 2.7. Rumen Fluid Sample Processing and Metabolite Profiling Analysis

Liquid chromatography-tandem mass spectrometry (LC-MS/MS) was used to determine rumen fluid metabolomics profiles. Chromatographic analysis was performed using the Waters Acquity ultra-high-performance system (1290 UHPLC; Agilent, Santa Clara, CA, USA), and nontargeted metabolite quantification was facilitated by high-resolution mass spectrometry (HRMS; TripleTOF 5600; AB Sciex, Framingham, MA, USA). Stool samples (approximately 50 mg) were triturated in precooled methanol (CNW Technologies, Dusseldorf, NW, GER) with a bead mill (TissueLyser; Qiagen, Beijing, China). The mixtures were then incubated for 10 min at 0 °C and 1 h at 220 °C before being centrifuged at 13,000 rpm at 4 °C for 15 min. The sample supernatant (20 mL) was collected and mixed with 200 mL of quality control (QC) material before being injected into the LC-MS/MS apparatus for further analysis.

The mass spectrometry data were collected using the AB 5600 TripleTOF mass spectrometer using Analyst software (Analyst TF 1.7; AB Sciex, Framingham, MA, USA). The time-of-flight (TOF) parameters were as follows: bombardment energy is 30 eV. The electrospray ionization (ESI) ion source parameters are as follows: atomization pressure (GS1) of 60 Psi, auxiliary pressure of 60 Psi, air curtain pressure of 35 Psi, temperature of 650 °C, and spray voltage of 5000 V (positive ion mode) or 24,000 V (negative ion mode). To eliminate chemical noise from raw LC-MS data, Genedata Expressionist software (v9.0) was used. Chromatographic peak identification, integration, normalization, and alignment retention durations between samples were all part of the process. After the data were analyzed, principal-component analysis (PCA) and orthogonal to partial least-squares discriminate analysis (OPLS-DA) were performed to visualize the group-to-group differences in metabolic profile. When combined, the substantially different metabolites were taken into account with a *p*-value of <0.05 and a variable influence on projection (VIP) of>1.

### 2.8. Data Analysis

Statistical analysis was performed using SPSS (v21.0) software. We used GraphPad Prism (9.0) software to draw a column chart. Student’s *t*-test was used for comparisons between two groups, and ANOVA was used for multi-group comparisons. *p* < 0.05 was considered statistically significant, and *p* < 0.01 was considered highly significant.

### 2.9. Co-Occurrence Network Analysis

We created a network of connections based on the relative abundance of each group to better understand the interactions between the bacteria in the rumen. In Python (3.11.4) software, Spearman’s correlation coefficient is used to analyze the associated networks in the CON and BR4 groups. The correlation between the genus was visualized using Cytoscape v3.0.1 (|Rho| > 0.60).

### 2.10. Correlation Analysis

We used the Scipy package in Python to compute the Spearman coefficient and the matplotlib package to visualize the results to examine the connection between rumen enzyme activity, serum oxidation parameters, metabolites, and microbes.

## 3. Results

### 3.1. The Impact of Different BR % and Ages on Serum Oxidative Indicators

To investigate the changes in serum oxidation indexes and the relationship between body weight and MDA during the growth process of SFWS, we compared the serum oxidation indexes of SFWS in the CON group at two age groups, 90 days of age (90 d) and 150 days of age (150 d), and analyzed the associations between body weight and MDA in the two age groups. Figure 1A reveals that the concentrations of T-AOC and MDA increased significantly (*p* < 0.01, *p* < 0.05) as the age of the SFWS grew after weaning. However, there was no significant variation in the contents of SOD and CAT, though they indicated an increasing tendency. The Spearman correlation coefficients between body weight and MDA were calculated and found to be significantly positive (*p* = 0.0227, rho = 0.71), as shown in Figure 1A. Figure 1B shows that increasing the proportion of BR led to higher concentrations of T-AOC and SOD in the BR4 group compared to the CON group (*p* < 0.01), while MDA concentrations decreased in the BR4 group (*p* < 0.05). The CAT concentration was not substantially different between the four groups, although the BR4 group showed an increase when compared to the CON group. The results showed that the concentration of MDA increased dramatically as the SFWS’ age and body weight increased, as did their T-AOC concentration, indicating an improvement in their antioxidant ability to cope with OS. Interestingly, feeding a 4% concentration of BR to SFWS increased their antioxidant capacity considerably. Thus, the BR4 and CON groups were chosen for follow-up in this study.

### 3.2. Impact of BR on Rumen Enzyme Activities

Figure 2 shows that the enzyme activities of amylase and pepsin in the BR4 group were significantly higher than those in the CON group (*p* < 0.05, *p* < 0.01). The enzyme activities of cellulase in the BR4 group showed an increasing trend, while the enzyme activity of lipase showed a decreasing trend. This suggested that BR had an influence on rumen enzyme activity in postweaning SFWS.

### 3.3. Impact of BR on Rumen Microbiota

Figure 3A shows that the Chao1 index of microbes in the BR4 group was significantly higher than that of the CON group based on the *α*-diversity of the microbiota (*p* < 0.05). However, there was no significant difference in the Shannon index. The PCoA analysis of the microbes reveals a significant difference in *β*-diversity between the BR4 and CON groups (Figure 3B). These findings indicate that BR influences the diversity of rumen bacteria. To evaluate the influence of BR on alterations to the composition of the rumen bacteria, changes in microbial composition at the phylum and genus levels were analyzed using Taxonomy. Figure 3C shows that *Firmicutes* and *Bacteroidota* were the most dominant phylum in both groups. In the BR4 group, the relative abundance of *Firmicutes* increased by 10.2% while *Bacteroidota* fell by 5.0% compared to the CON group. Figure 3D shows that the relative abundance of *Prevotella* in the BR4 group reduced by 5.8%, while *Ruminococcus* increased by 5.9% compared to the CON group at the genus level. This suggests that BR increases the diversity of rumen microbiota in post-weaned SFWS and also influences the compositional structure of their rumen bacteria.

Next, we used Metastat to conduct a differential relative abundance analysis to accurately investigate the microbes in different groups. The relative abundance of the different rumen microbiota varied between the CON and BR4 groups at the genus and species levels. At the genus level, *Marvinbryantia* and *Methanosphaera* were considerably more abundant in the BR4 group compared to the CON group (*p* < 0.01, *p* < 0.05) (Figure 4A). The relative abundance of *Ruminococcus* increased in the BR4 group, while *Succiniclasticum* dropped (Figure 4A). In comparison to the CON group, the relative abundance of *Ruminococcus_sp_N15_MGS_57* and *Ruminococcus*_*sp* increased in the BR4 group, but *Ruminococcus_albus* and *Ruminococcus_flavefaciens* decreased (Figure 4B). The findings suggest that the microbiota’s major components remain independent, with some overlap between the CON and BR4 groups. At the same time, feeding with 4% BR resulted in a distinct rumen microbial composition dominated by a diverse range of functional beneficial bacteria.

### 3.4. The Impact of BR on the Coexistence Patterns of Rumen Microbes

Microbial co-occurrence networks have been commonly used to investigate inter-microbial relationships in rumen microbes and microbial interactions in the networks are related to differing microbial competitive strengths or ecological niche differentiation. Using Python, we calculated the relationship between the TOP 30 genera at the genus level between the CON and BR4 groups, screened the groups with absolute correlation coefficients greater than 0.6, and used Cytospace to construct the two groups’ microbial co-occurrence networks. Table 2 shows that the CON group has more nodes (*n* = 19) and edges (*n* = 122) than the BR4 group, and the average clustering coefficient (0.64) is greater than the BR4 group’s (0.37). This revealed that the CON group had a more complicated microbial co-occurrence network than the BR4 group. Vertex weighting calculated the relationships between edges and nodes in the network to screen the core group. As shown in Figure 5A, the core group of the CON group was screened with eight bacteria: *unidentified_Oscillospiraceae*, *Anaerovibrio*, *Achromobacter*, *Ruminobacter*, *Alloprevotella*, *Ruminococcus*, *unidentified_Rhodospirillales*, and *Succiniclasticum*. Figure 5B shows that five core bacteria were tested in the BR4 group, including *Butyrivibrio*, *Anaerovibrio*, *Ruminobacter*, *Methanobrevibacter*, and *Selenomonas.* These findings suggest that the core bacteria differed significantly between the two groups.

### 3.5. Impact of BR on Rumen Metabolites

We conducted rumen metabolomic studies on CON and BR4. The findings of the partial least squares discriminant analysis in positive and negative ionization modes revealed that CON and BR4 had distinct metabolic groups (Figure 6A,B). The volcano figure shows that in the positive ionization mode, 175 metabolites were up-regulated and 169 down-regulated (Figure 6C); in the negative ionization mode, 205 metabolites were up-regulated and 47 down-regulated (Figure 6E). Subsequent screening of the TOP 20 significantly different metabolites for analysis showed that, in comparison to the CON group, there was a notable up-regulation of spirolide C and 5,6-DHET and a significant down-regulation of oleamide, PE-NMe(15:0/22:5(7Z,10Z,13Z,16Z,19Z)), 1, 2-Dimyristoyl-sn-glycero-3-phosphocholine,1-eicosanoyl-2-(13Z,16Z-docosadienoyl)-glycero-3-phosphate, and LPA (a-25:0/0:0) (Table 3). To determine whether the differential metabolites in one specific path were significantly different, the KEGG pathway enrichment analysis was applied. The top 20 positive ionization mode pathways are displayed in Figure 6E. The metabolic pathways are the most significantly enriched pathways, followed by the metabolism of glycerophospholipids, lipids, and inositol phosphates, among others. Figure 6F displays the top 20 pathways of TOP in the negative ionization mode. The metabolic pathway is the most significantly enriched pathway, followed by Pyrimidine metabolism, Glutathione metabolism, Histidine metabolism, Ascorbate and aldarate metabolism, 2–Oxocarboxylic acid metabolism, fatty acids metabolism, and D-amino acids metabolism.

### 3.6. Spearman Correlation Analysis

This study evaluated the relationships between rumen microbial diversity and OS, as well as the relationships between rumen microbes and enzyme activities, OS, and metabolites. Specifically, Spearman analysis was used to examine the relationships between the TOP 10 rumen microbiota and OS, as well as the relationships between the TOP 20 metabolites and OS. Certain bacteria were discovered to have positive or negative correlations with OS and enzyme activity (Figure 7A,B). As an illustration, consider the following: 5,6-DHET (MW0014628) was significantly positively correlated with MDA, while LPA (a-25:0/0:0) (MW0054512) was significantly negatively correlated with MDA. *Ruminococcus* was significantly positively correlated with amylase, and *Prevotella* was significantly negatively correlated with cellulase. SOD was negatively correlated with *Prevotella*, and T-AOC was significantly positively correlated with *unidentified_Bacteroidales* (Figure 7A,C). Finally, using Spearman analysis to explore the association between OS and rumen microbiota diversity, we discovered a striking negative correlation between the Chao1 index and MDA, but a positive correlation with SOD and T-AOC (Figure 7B).

## 4. Discussion

Improving the antioxidant capacity of SFWS that have been weaned is an essential approach to enhancing their overall health. Research on the impact of rumen bacteria on host physiological processes has mostly studied growth and development, as well as digestion and absorption, with less attention paid to the effects on OS. Notably, it has been demonstrated recently that elevated host OS may be a harmful factor in certain diseases resulting from microbial disorders [26,27,28,29]. Microbiota and OS are therefore related, and they are crucial for preserving the healthy condition of the host, the dynamic balance of metabolism, and the digestion and absorption of nutrients. Our research indicates that by adjusting the rumen microbiota of SFWS and their metabolites, which further affect the activity of antioxidant enzymes, BR can enhance host antioxidant capacity and lessen OS.

Lamb’s OS is impacted by many factors, including host genetics, illnesses, and various environments [30,31,32]. OS and aging are directly correlated, according to an increasing number of studies [33,34,35]. Lambs were shown to have much higher levels of oxidative damage than lambs of other ages in research on oxidative damage in lamb, and these levels were also found to be significantly higher throughout the first four months of life [36]. For four to five months following birth, growth rates in lambs are related to higher blood MDA concentrations [36]. According to the current study, body weight significantly and positively correlated with MDA, and the concentration of MDA increased significantly with increasing day of age in weaned SFWS between 90 and 150 days of age. These findings are consistent with the research of Nussey et al. [36]. In the current study, it was discovered that, although SOD and CAT contents did not change considerably but did show a rising tendency, the T-AOC concentration increased significantly as the age of the SFWS increased. Thus, we hypothesized that the rise in T-AOC in SFWS as they become older offsets the rise in OS levels. Adding several relevant antioxidant compounds to the diets has been used to increase the antioxidant capacity and overall health of sheep. For instance, Liang et al. [37] discovered that dietary supplementation with spirulina alleviated OS related to high-energy diets, while Kafantaris et al. [38] fed lambs grape pomace, a by-product of winemaking, and discovered that OS in lambs was decreased by grape pomace. BR is a popular Chinese herb that contains biological properties like antioxidant activity [39]. In comparison to the control, Bai et al. [17] observed that adding BR to the diet dramatically decreased the MDA levels and greatly raised the SOD and CAT enzyme activities in the liver of broiler chickens. In the current study, we explored the impact of differing BR feeding proportions on OS in SFWS. Our findings indicated that the levels of T-AOC and SOD content increased as BR increased, with the BR4 group showing a significantly higher concentration than the CON group. The MDA concentration decreased as BR increased, with the BR4 group showing a significantly lower concentration than the CON group. The CAT content did not differ statistically between the four groups, but there was a trend for the BR4 group’s level to rise in comparison to the CON group’s. This indicates that in post-weaning SFWS, BR considerably increases antioxidant capacity and decreases OS.

The microbiota diversity, composition, and several significantly distinct genera between the BR4 and CON groups were found to differ in this study. The Chao1 index showed a significant negative relationship with MDA and a positive correlation with SOD and T-AOC. The Chao1 index was significantly higher in the BR4 group than in the CON group, whereas the Shannon index did not differ significantly. The greater the richness and diversity of the microbiota in the sample, the higher the Chao1 index and Shannon index [40]. Hence, we hypothesized that BR might affect the diversity and abundance of rumen microbes in SFWS, increasing their antioxidant levels and lowering OS. Additionally, comparing the BR4 group to the CON group, the relative abundance of *Bacteroidota* dropped by 5.0% and *Firmicutes* increased by 10.2% at the phylum level. While *Firmicutes*, the primary phylum of the rumen in ruminants, is mostly composed of a variety of fiber-degrading bacteria and cellulolytic bacteria of the genus, *Bacteropoda* plays a role in the digestion of carbohydrates [41,42]. The current research found that the ratio of *Firmicutes* to *Bacteroidota*, which has been used as an essential indicator for assessing the effect of microbes on the energy needs of the host, was higher in the BR4 group than in the CON group. This implies that by improving SFWS’ energy consumption, BR may be able to lower energy needs. *Marvinbryantia*’s relative abundance at the genus level was considerably higher in the BR4 group than in the CON group. Previous research has demonstrated that intestinal microbial disorders in colitis-affected mice can be prevented by microbiome-derived inosine. This is achieved by both decreasing the abundance of pathogenic bacteria (*Pseudomonas*, *Acinetobacter*, and *Tyzzerella*) and increasing the abundance of beneficial bacteria (*Lachnospiraceae NK4A136 group*, *Romboutsia*, *Marvinbryantia*, and *Bifidobacterium*) [43]. Therefore, we speculated that BR could intervene to boost the proportional abundance of advantageous bacteria to intervene in rumen microbiota disturbance and preserve the health of SFWS. Research indicates that supplementing with niacin can potentially mitigate heat stress by elevating the relative abundance of *Succiniclasticum* [44]. According to the majority of research, OS and heat stress are tightly related [45,46,47,48]. Given the current study’s findings—which included an increase in *Succiniclasticum*’s relative abundance at the genus level in the BR4 group and a significant negative correlation between *Succiniclasticum* and T-AOC—we postulated that increasing *Succiniclasticum*’s relative abundance would improve SFWS’ antioxidant capacity. Numerous studies have discovered a negative correlation between increased OS and several diseases and the quantity of the beneficial bacterium *Ruminococcus* [49,50,51]. According to Yang et al. [52], mice’s hepatic OS was lowered when *Rumenococcaceae* abundance increased. Gu et al. [26] discovered that low-OS cow groups had higher relative abundances of bacterial taxa belonging to the *Ruminococcaceae* family, such as *R. bacterium*, *R. flavefaciens*, and *R. CAG:724*. Consistent with these studies, *Ruminococcus_sp_N15_MGS_57* and *Ruminococcus_sp* showed higher relative abundance in the BR4 group. Also, this study discovered a positive correlation between *Ruminococcus* and T-AOC as well as an increase in the relative abundance of *Ruminococcu* at the genus level in the BR4 group. Hence, by influencing alterations in these microbiota, BR may have a significant impact on host OS. In addition, a multitude of enzymes that are crucial to the digestive process are secreted and produced by these bacteria. The activities of these enzymes are primarily determined by the species and quantity of rumen microorganisms and are intimately related to their metabolic processes [53]. Rumen microbes were found to be closely related to rumen enzyme activities in a study by Moharrery et al. [54] that examined the correlation of microbial enzyme activities in sheep’s rumen fluid under various treatments. Similarly, in the current study, we discovered that *Ruminococcus* was significantly and positively correlated with amylase and *Prevotella* was significantly and negatively correlated with cellulase. This is consistent with Moharrery et al.’s findings. By generating amylase, *Ruminococcus* are microorganisms that may ferment starch and other carbohydrates in the rumen [55,56]. The rumen environment affects amylase activity, and the content and intake of the diet also have an impact on the quantity and activity of *Ruminococcus* and amylase [57]. Several rumen bacteria, such as *Butyrivibrio fibrisolvens* and *F. succinogene*, have been discovered in some research to form macromolecular complexes comprising multiple cellulases [58]. Interestingly, the current study demonstrated a significant negative correlation between *Prevotella* and cellulase. In addition, we found that Pepsin was significantly positively correlated with various metabolites such as 1,2-Dioleoyl-sn-glycero-3-phosphoethanolamine, TG (15:0/15:0/22:5(4Z,7Z,10Z,13Z,16Z)) and [(2R)-2-(15-methylhexadecanoyloxy)-2-(15-phosphonooxypropyl] icosanoate were significantly positively correlated with various metabolites, and amylase was significantly negatively correlated with 5,6-DHET. Thus, we postulated a strong relationship between metabolites and rumen microorganisms and enzyme activity.

Studies on the association between the rumen microbiota and OS are still underreported, and the mechanisms underlying these findings are not well understood, despite the general interest in the effects of the gut microbiota on host OS. Crucial cellular processes like energy synthesis and storage are regulated by metabolites and have an impact on rumen condition. Thus, using metabolomics approaches, this study analyzed the impact of BR supplementation on SFWS’ rumen metabolomics. It has been discovered that 5,6-DHET decreases macrophages’ synthesis of IL-6 [59]. It is believed that IL-6 could reduce host OS and play a role in the initiation and development of numerous illnesses, including inflammation. On the other hand, we discovered that 5,6-DHET was markedly elevated in the BR4 group, indicating that a rise in 5,6-DHET might reduce OS. All tissues contain lysophosphatidic acid (LPA), a physiologically active lysophospholipid that causes inflammation and OS [60]. The present study revealed that LPA(a-25:0/0:0) was considerably down-regulated in the BR4 group, indicating that decreasing LPA(a-25:0/0:0) may lessen the probability that OS will occur in the host. The key metabolic pathways enriched in positive ionization mode, according to KEGG pathway analysis, were inositol phosphate metabolism, glycerophospholipid metabolism, and glycerolipid metabolism. The glycerophospholipid metabolism pathway is essential for OS, as numerous studies have demonstrated [61,62,63]. Meanwhile, Zhu et al. [64] showed that three OS products (SOD, MPO, and 8-iso-PGF2α) and glycerophospholipid metabolites had a significant negative connection. According to Mo et al. [65], thiram may interfere with the metabolism of taurine and hypotaurine, pyrimidines, and glycerolipids, which in turn may cause OS by activating the Nrf2 signal pathway. Astaxanthin has been shown by Mei et al. [66] to partially counteract the down-regulation of specific metabolic pathways, including glycerolipid metabolism, and reduce OS and cell death. In the lungs of rats modeled with sepsis-related lung injury, He et al. [67] discovered that Liang-Ge decoction inhibited inflammatory responses, OS, apoptosis, and modulated metabolites associated with glycine, serine, and threonine metabolism, cysteine and methionine metabolism, inositol phosphate metabolism, and the TCA cycle. Pyrimidine metabolism, 2-oxocarboxylic acid metabolism, nucleotide metabolism, and D-amino acid metabolism are the primary metabolic pathways that are enriched in the negative ionization mode. Through studying the cytoprotective impact of lignans on glutamate-induced OS injury in PC12 cells, it was discovered that lignans’ ability to regulate amino acid, glucose, and nucleotide metabolic pathways is linked to their systemic antioxidant capacity in PC12 cells [68]. Different microbial metabolic patterns of amino acids, including those related to the metabolism of glutamine and glutamate, glycine, and cysteine, have been proposed as potential contributors to host OS [26]. Therefore, we postulated that by influencing the metabolism of glycerolipids, glutathione, nucleotides, D-amino acids, and inositol phosphate, BR supplementation could enhance antioxidant capacity and lessen OS in SFWS.

## 5. Conclusions

To summarize, our study reveals that BR may reduce OS and boost the host’s antioxidant potential by adjusting the diversity and composition of rumen microbes and metabolites in SFWS. Key rumen microflora included *Succiniclasticum* and *Ruminococcus* at the genus level and *Ruminococcus_sp_N15_MGS_57* and *Ruminococcus_sp* at the species level, while the key metabolites were 5,6-DHET and LPA (a-25:0/0:0). Furthermore, the BR4 group’s high antioxidant activity and decreased OS were related to BR’s ability to regulate the metabolism of glycerolipids, glutathione, nucleotides, D-amino acids, and inositol phosphate. Our results suggest that appropriate feeding plans will offer new perspectives on improving health during the growth of SFWS.

## Figures and Tables

**Figure 1 animals-14-00927-f001:**
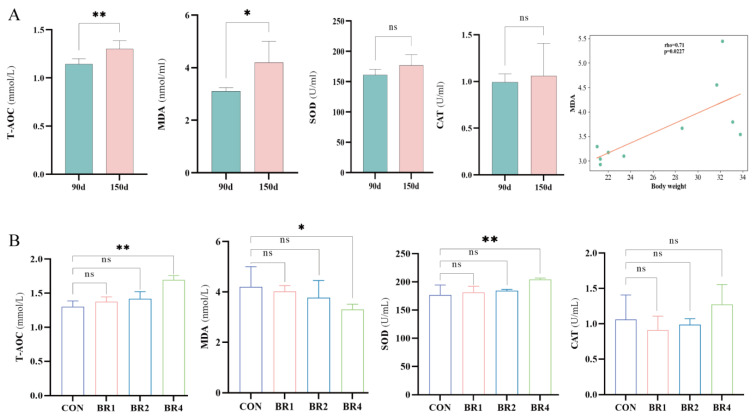
Comparison of the effects of different BR % and age on serum oxidative indices and analysis of the relationship between MDA and body weight. (**A**) Effect of age on serum oxidative indices and analysis of the relationship between MDA and body weight. (**B**) Effect of different BR % on serum oxidative indices. * denotes significant, ** denotes highly significant, ns denotes insignificant.

**Figure 2 animals-14-00927-f002:**
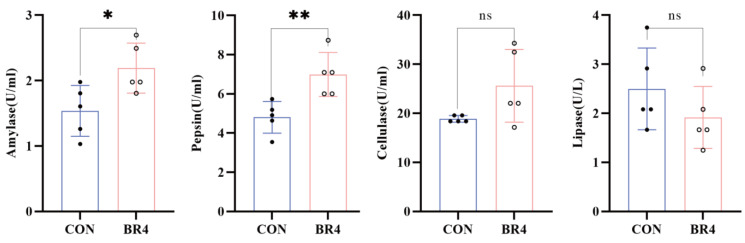
Effect of BR on rumen enzyme activity. * denotes significant, ** denotes highly significant, ns denotes insignificant.

**Figure 3 animals-14-00927-f003:**
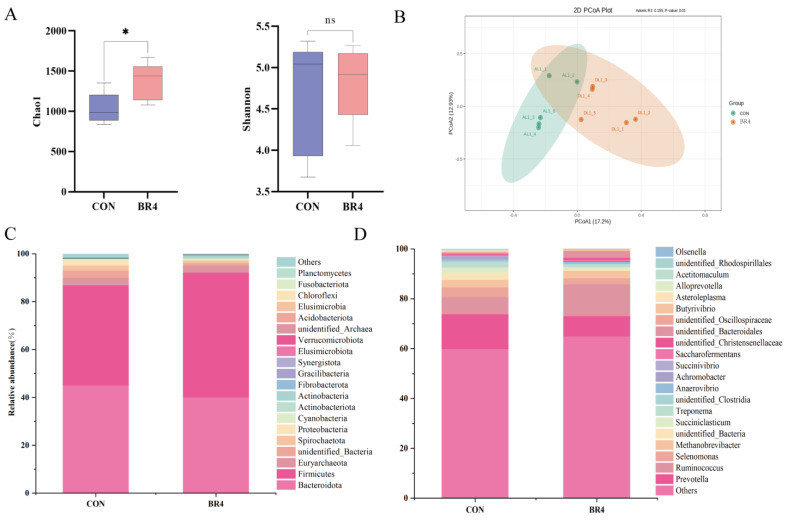
Effect of BR on rumen microbial composition and diversity. (**A**) Chao1 and Shannon index. (**B**) PcoA. (**C**) Changes in the composition of rumen microbiota at the phylum level. (**D**) Changes in the composition of rumen microbiota at the genus level. * denotes significant, ns denotes insignificant.

**Figure 4 animals-14-00927-f004:**
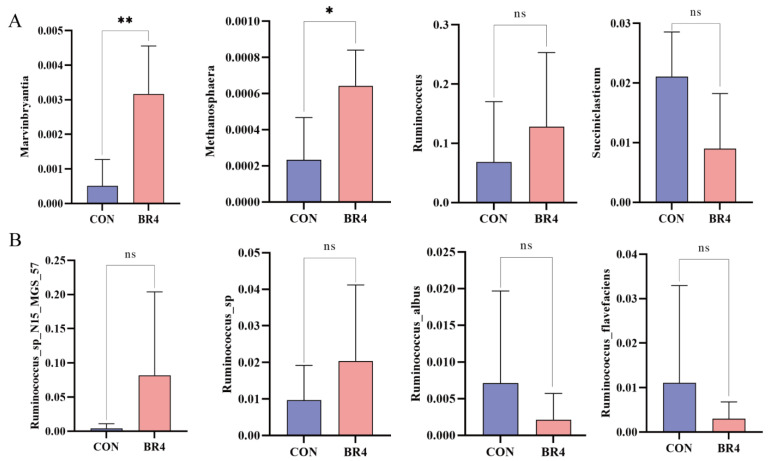
Significant differences in the relative abundance of rumen microbiota between the two groups were observed at the genus and species levels. (**A**) genus level. (**B**) species level. * denotes significant, ** denotes highly significant, ns denotes insignificant.

**Figure 5 animals-14-00927-f005:**
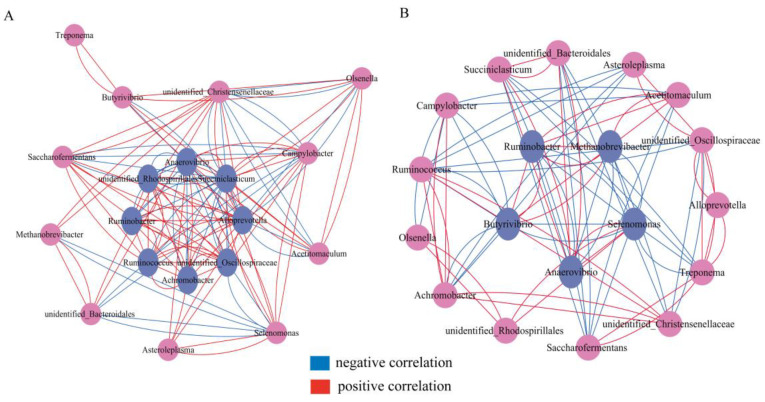
Genus co-occurrence network between CON and BR4 based on Spearman correlation analysis. (**A**) CON group. (**B**) BR4 group. Each node represents a bacterial genus. The line refers to the Spearman coefficient. Red and light blue lines represent positive and negative interactions between nodes, respectively. Correlations with |rho| > 0.6 are presented.

**Figure 6 animals-14-00927-f006:**
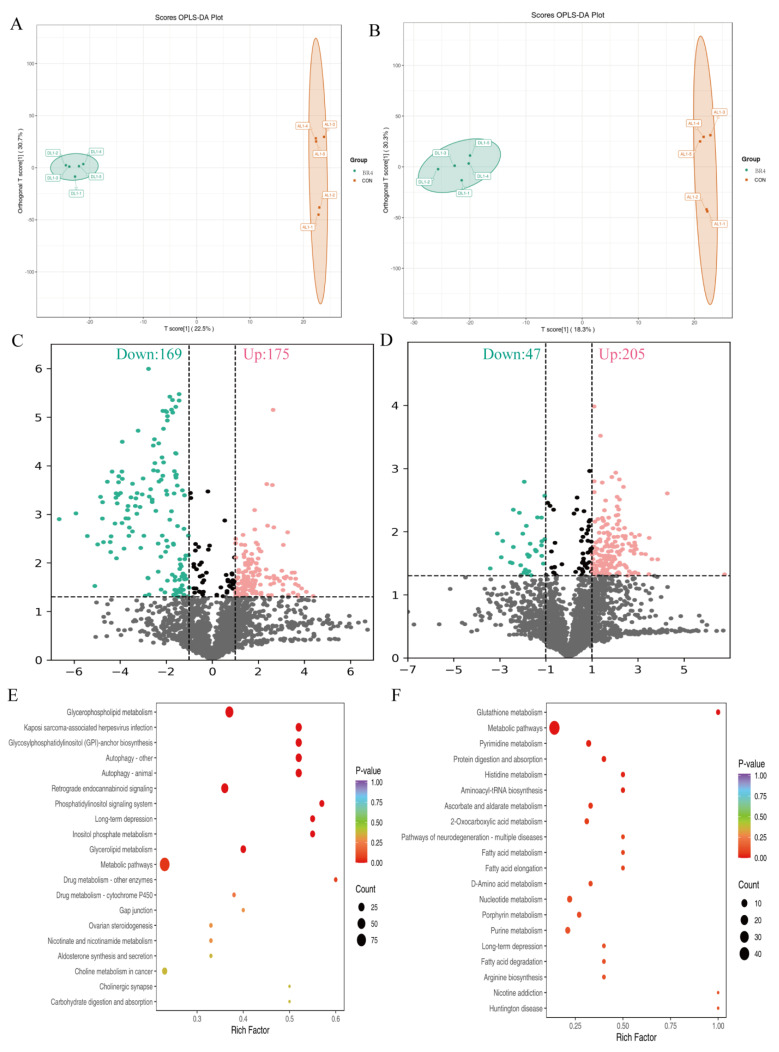
Different metabolic patterns between the two groups. (**A**) OPLS-DA score plot of rumen metabolome in positive ionization mode. (**B**) OPLS-DA score plot of rumen metabolome in negative ionization mode. (**C**) Volcano map of rumen metabolome in positive ionization mode. (**D**) Volcano map of rumen metabolome in negative ionization mode. (**E**) KEGG pathway map of the rumen metabolome in positive ionization mode. (**F**) KEGG pathway map of the rumen metabolome in negative ionization mode.

**Figure 7 animals-14-00927-f007:**
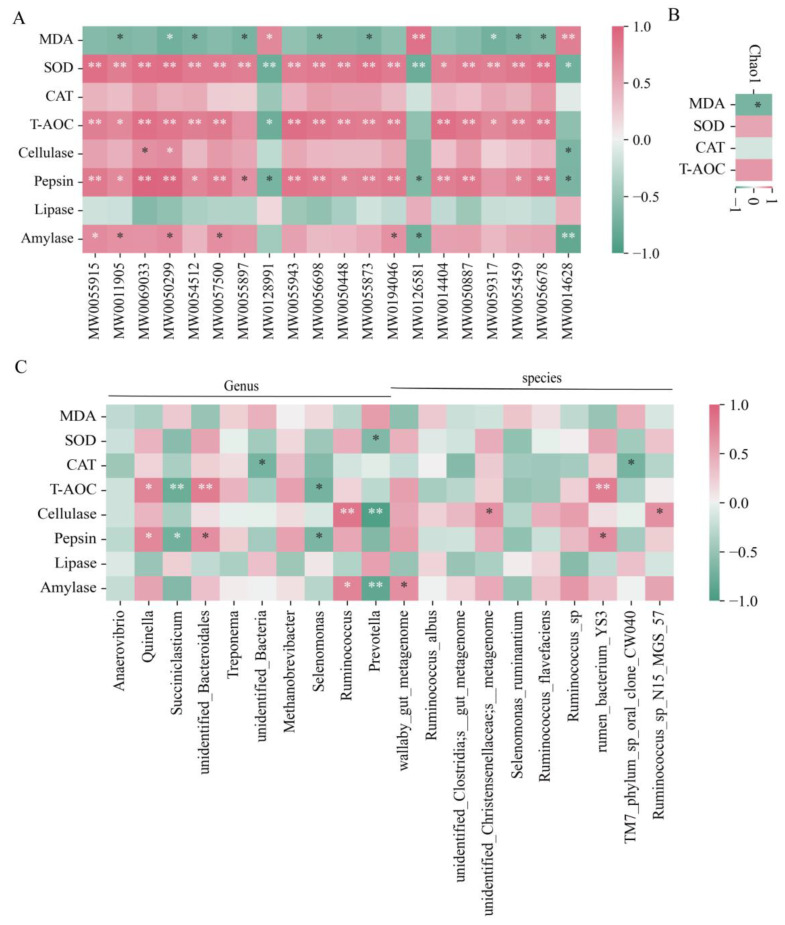
Spearman correlation analysis. (**A**) Association analysis of rumen metabolites with serum oxidative indices and enzyme activities. (**B**) Association analysis of microbial diversity with serum oxidative indices. (**C**) Association analysis of rumen microbiota with serum oxidative indices and enzyme activities. * denotes significant, ** denotes highly significant.

**Table 1 animals-14-00927-t001:** Basic diet composition and nutrient levels.

Ingredients	Content
Corn	28.85
Soybean meal	7.50
Premix	3.20
Alfalfa grass	22.45
Corn for silage	38.00
Total	100
Nutrient levels (%)	
Crude protein	18.91
Crude fat	1.80
Crude fiber	10.79
Crude ash	3.70
Ca	0.32
P	0.15
Lysine	≥0.4
Digested energy	11.13%

**Table 2 animals-14-00927-t002:** Nodes, number of edges, and average clustering coefficient between two groups.

	Index	Node	Edge	Average Clustering Coefficient
Group				
CON		19	122	0.64
BR4		19	80	0.37

**Table 3 animals-14-00927-t003:** Significant differences between the two groups’ TOP 20 metabolites.

Index	Compounds	log_2_FC	Type
MW0056698	[(2R)-3-phosphonooxy-2-tetradecanoyloxypropyl] 22-methyltricosanoate	−6.64	down
MW0050887	1-docosanoyl-2-(4Z,7Z,10Z,13Z,16Z,19Z-docosahexaenoyl)-sn-glycerol	−5.91	down
MW0055943	[(2R)-1-[(Z)-icos-11-enoyl]oxy-3-phosphonooxypropan-2-yl] (Z)-docos-13-enoate	−5.42	down
MW0059317	PE-NMe(15:0/22:5(7Z,10Z,13Z,16Z,19Z))	−5.10	down
MW0055459	Oleamide	−4.97	down
MW0056678	[(2R)-2-(10-methyldodecanoyloxy)-3-phosphonooxypropyl] 20-methylhenicosanoate	−4.85	down
MW0011905	1,2-Dioleoyl-sn-glycero-3-phosphoethanolamine	−4.77	down
MW0054512	LPA(a-25:0/0:0)	−4.75	down
MW0069033	TG(15:0/15:0/22:5(4Z,7Z,10Z,13Z,16Z))	−4.72	down
MW0014404	4alpha-Formyl-4beta-methyl-5alpha-cholesta-8,24-dien-3beta-ol	−4.55	down
MW0050448	DG(20:0/22:5(4Z,7Z,10Z,13Z,16Z)/0:0)	−4.43	down
MW0057500	PC(22:1(13Z)/22:5(4Z,7Z,10Z,13Z,16Z))	−4.41	down
MW0050299	DG(18:4(6Z,9Z,12Z,15Z)/20:3(5Z,8Z,11Z)/0:0)	−4.39	down
MW0194046	1,2-Dimyristoyl-sn-glycero-3-phosphocholine	−4.35	down
MW0055873	1-eicosanoyl-2-(9Z-tetradecenoyl)-glycero-3-phosphate	−4.22	down
MW0055915	[(2R)-2-(15-methylhexadecanoyloxy)-3-phosphonooxypropyl] icosanoate	−4.18	down
MW0055897	1-eicosanoyl-2-(13Z,16Z-docosadienoyl)-glycero-3-phosphate	−4.17	down
MW0128991	(5-{8-[1-(2,4-dihydroxyphenyl)-3-(3,4-dihydroxyphenyl)-2-hydroxypropyl]-3,5,7-trihydroxy-3,4-dihydro-2H-1-benzopyran-2-yl}-2-hydroxyphenyl)oxidanesulfonic acid	4.28	up
MW0014628	5,6-DHET	4.39	up
MW0126581	spirolide C	6.76	up

## Data Availability

All data in this study are available upon request by contact with the corresponding author. The datasets generated for this study can be found in the NCBI Sequence Read Archive under BioProject PRJNA1067348.

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
