# Peer review of "Dietary Supplementation with Bupleuri Radix Reduces Oxidative Stress Occurring during Growth by Regulating Rumen Microbes and Metabolites"

_animals, 2024, doi:10.3390/ani14060927_

Round 1

Reviewer 1 Report

Comments and Suggestions for Authors

I extend my gratitude for the opportunity to review the manuscript titled "Dietary supplementation with Bupleuri Radix reduces oxidative stress occurring during growth by regulating rumen microbes and metabolites." Overall, I find the study to be meticulously conducted and deserving of publication consideration.

The findings of this study are significant, demonstrating that Bupleuri Radix (BR) supplementation may effectively mitigate oxidative stress and enhance the antioxidant capacity of the host by modulating the diversity and composition of rumen microbes and metabolites in Shanbei Fine Wool Sheep (SFWS). Notable rumen microflora identified include Succiniclasticum and Ruminococcus at the genus level, and Ruminococcus_sp_N15_MGS_57 and Ruminococcus_sp at the species level. Key metabolites such as 5,6-DHET and LPA (a-25:0/0:0) were also identified. Furthermore, the observed high antioxidant activity and reduced oxidative stress in the BR-treated group appear to be associated with the modulation of glycerolipid, glutathione, nucleotide, D-amino acid, and inositol phosphate metabolism.

The implications of these findings suggest that implementing appropriate dietary strategies could offer novel insights into improving the health and well-being of growing SFWS populations.

The manuscript is well-structured, with a clear and informative introduction that effectively sets the stage for the study. The purpose and scope of the research are adequately delineated, and the methodology is well-described, enabling reproducibility of the experiments. The presentation of results and subsequent discussion are coherent and logically organized.

While the manuscript is generally sound, there are some minor revisions that should be addressed prior to acceptance:

  1. Line 22-23, In the Abstract section, ensure consistency in capitalization for Malondialdehyde (MDA) and Superoxide Dismutase (SOD) throughout the manuscript.
  2. Line 46 and Line 56: Provide the full name of the abbreviation "BR" at its first mention in the Introduction section, and similarly for "SFWS" in subsequent mentions.
  3. Include the Ethical code number in the Ethical statement in the Materials and Methods section (Lines 79-83).
  4. Correct the title of the reference and rewrite again listed in Lines 600-601 to read: "Metabolomics study based on GC-MS reveals a protective function of luteolin against glutamate-induced PC12 cell injury."

Gao, Ying, et al. "Metabolomics study based on GC–MS reveals a protective function of luteolin against glutamate‐induced PC12 cell injury." Biomedical Chromatography 37.2 (2023): e5537.

Overall, I find the manuscript to be engaging and valuable to the scientific community. With the suggested revisions, I believe it will be a strong addition to the literature.

Comments on the Quality of English Language

I think a few minor edits will be required.

Author Response

Dear reviewer:

We feel great thanks for your professional review work on our article. These comments were of great help to our article. According to your nice suggestions, we have made extensive corrections to our previous draft, the detailed corrections are listed below.

  1.    Line 22-23, In the Abstract section, ensure consistency in capitalization for Malondialdehyde (MDA) and Superoxide Dismutase (SOD) throughout the manuscript.

A: We feel great thanks for your professional review work on our article. We have modified the capitalization of Malondialdehyde (MDA) and Superoxide Dismutase (SOD) in lines 32-33

  1.    Line 46 and Line 56:Provide the full name of the abbreviation "BR" at its first mention in the Introduction section, and similarly for "SFWS" in subsequent mentions.

A: We have supplemented all first mentions of abbreviations in the introduction section with their full names.(Line 46/56/67)

  1.    Include the Ethical code number in the Ethical statement in the Materials and Methods section (Lines 79-83).

A: We have added the Ethical code number to the ethics statement in the Materials and Methods section.(Line 94)

  1.    Correct the title of the reference and rewrite again listed inLines 600-601to read: "Metabolomics study based on GC-MS reveals a protective function of luteolin against glutamate-induced PC12 cell injury."

A: We have revised the title of this literature .(Line 650)

Reviewer 2 Report

Comments and Suggestions for Authors

Brief Summary

The article describes the impact of Bupleuri Radix on Shanbei Fine Wool Sheep, associating rumen parameters, microbiota investigation and transcriptomic to identify Key biomarkers. The article is written in very good English, and the material and methods used seem adequate as far as I can judge. The presentation of the many results as the discussion are good and request only minor changes.

General Concept

The article is very well structured, the results are well presented, coherent and the discussion convincing. My main only point is that for me it remains unclear is the very positive results recorded and discussed, are linked to a direct effect of BR or to the degradation of the product within the rumen. If the authors should add some comments on that regard, it would be perfect.

Comments on the Quality of English Language

Specific comments referring to line numbers,

Line 182 : “conducting” please improve english ?

Line 182 : figure 1 : vertical legend SOD U/ml is not fully visible.

Author Response

Dear reviewer:

We feel great thanks for your professional review work on our article. These comments were of great help to our article. According to your nice suggestions, we have made extensive corrections to our previous draft, the detailed corrections are listed below.

  1.    Line 182 : “conducting” please improve english ?

A: We've corrected the language problem in Line182.(Line 223)

  1.    Line 182 : figure 1 : vertical legend SOD U/ml is not fully visible.

A: We have modified and adjusted all the images in Figure 1.(Line 226)

Reviewer 3 Report

Comments and Suggestions for Authors

This ms still needs to clarify much of ackwardness in the manuscript preparation to meet with good soundness of their research.

L84- Need much more info about:

- the age period of animals who received the test, D90 - 150?

- composition and feeding amount of their basal diet

- also needs the nutrition spec for BR

L176 Why the are talking about methane here? - Rename the grouping abbreviations in the text and Figures to such as BR1, 2, 4, for instance 

Figure1 Specify the grouping of data in FigA -- Was both CON and BR included in the dataset?

L190- (and further dataset in all Figs) how did they treat data from 10 animals to 5 plots per group?

Figure 3 The shape is ugly in Fig3C, reduce the width of each bar so as to ease readability of the caption.

* Detail method information about their metabolomics analysis must be provided to deal with data shown in Fig6

Comments on the Quality of English Language

Many awkwardness found through main text, starting with abbreviations or with numeric, and irregular spacing elsewhere, all of which indicate this ms needs to receive intensive English editing. 

Author Response

Dear reviewer:

Thank you for your positive comments and valuable suggestions to improve the quality of our manuscript. According to the comments, we have made extensive modifications to our manuscript and supplemented extra data to make our results convincing.

  1.   L84- Need much more info about:

- the age period of animals who received the test, D90 - 150?

- composition and feeding amount of their basal diet

- also needs the nutrition spec for BR

A: All test animals selected for this experiment were 3 months old, which is mentioned in the Materials and Methods section 2.2 of the article. D90 and D150 represent the two age groups of SFWS in the CON group, which are 90 and 150 days old, respectively. Changes in serum oxidative indices in these two age groups were examined to determine whether the oxidative stress experienced by SFWS is increased during the growing phase.

The composition of the basal diets fed in this experiment has been supplemented in the article, please refer to Table 1.( Line 116)

BR was used as an additional additive for this test, which was reflected in the article title. And, with reviewing and studying some relevant literature, we believe that BR as an additional additive can be used without measurining its nutrient content. The relevant literature is as follows:

  1. Shao P, Sha Y, Liu X, et al. Supplementation with Astragalus Root Powder Promotes Rumen Microbiota Density and Metabolome Interactions in Lambs[J]. Animals, 2024, 14(5): 788.
  2. Chen G, Li Z, Liu S, et al. Fermented Chinese herbal medicine promoted growth performance, intestinal health, and regulated bacterial microbiota of weaned piglets[J]. Animals, 2023, 13(3): 476.
  3. Wang C, Zhong Y, Liu H, et al. Effects of Dietary Supplementation with Tea Residue on Growth Performance, Digestibility, and Diarrhea in Piglets[J]. Animals, 2024, 14(4): 584.
  4.    L176 Why the are talking about methane here? - Rename the grouping abbreviations in the text and Figures to such as BR1, 2, 4, for instance.

A: We've renamed all the images for the article.

  1.    Figure1 Specify the grouping of data in FigA -- Was both CON and BR included in the dataset?

A: The main purpose of the two groups in Fig. 1A, all of which are from the CON group, was to investigate whether oxidative stress increased during the growth of SFWS. We also provide additional details in the text.( Line 207-211)

  1.    L190- (and further dataset in all Figs) how did they treat data from 10 animals to 5 plots per group?

A: In this experiment, at the end of the experiment, we slaughtered five rams in each group. The main reasons were as follows:

This research conducted more than only the contents described in this paper. We also detected the effects of BR feeding on the meat quality of SFWS and some other research work. In addition, 5 samples per group satisfy biological replication requirements. Therefore, we selected five rams in each group for slaughter and sampling collection.

  1.    Figure 3 The shape is ugly in Fig3C, reduce the width of each bar so as to ease readability of the caption.

A: We have made changes to Figures 3C-D.(Line 268)

  1.    * Detail method information about their metabolomics analysis must be provided to deal with data shown in Fig6.

A: Analysis methods for metabolomics have been supplemented in the Materials and Methods 2.7 section.(Line 166-189)
